# GeONet: a neural operator for learning the Wasserstein geodesic

**Andrew Gracyk**[1]                                **Xiaohui Chen**[2]

[1]Department of Statistics, University of Illinois at Urbana-Champaign
[2]Department of Mathematics, University of Southern California

## Abstract

Optimal transport (OT) offers a versatile framework to compare complex data distributions in a geometrically meaningful way. Traditional methods for computing the Wasserstein distance and geodesic between probability measures require mesh-specific domain discretization and suffer from the curse-of-dimensionality. We present *GeONet*, a mesh-invariant deep neural operator network that learns the non-linear mapping from the input pair of initial and terminal distributions to the Wasserstein geodesic connecting the two endpoint distributions. In the offline training stage, GeONet learns the saddle point optimality conditions for the dynamic formulation of the OT problem in the primal and dual spaces that are characterized by a coupled PDE system. The subsequent inference stage is instantaneous and can be deployed for real-time predictions in the online learning setting. We demonstrate that GeONet achieves comparable testing accuracy to the standard OT solvers on simulation examples and the MNIST dataset with considerably reduced inference-stage computational cost by orders of magnitude.

## 1 INTRODUCTION

Recent years have seen tremendous progress in statistical and computational optimal transport (OT) as a lens to explore machine learning problems. One prominent example is to use the Wasserstein distance to compare data distributions in a geometrically meaningful way, which has found various applications, such as in generative models [Arjovsky et al., 2017], domain adaptation [Courty et al., 2017] and computational geometry [Solomon et al., 2015]. Computing the optimal transport map (if it exists) can be expressed in a fluid dynamics formulation with the minimum kinetic energy [Benamou and Brenier, 2000]. Such a dynamical formulation defines geodesics in the Wasserstein space of probability measures, thus providing richer information for interpolating between data distributions that can be used to design efficient sampling methods from high-dimensional distributions [Finlay et al., 2020]. Moreover, learning the continuous-time dynamical Wasserstein geodesic is a practically important task in many science and engineering domains, including developmental trajectory reconstruction in cell reprogramming [Schiebinger et al., 2019], 3D warping for shape analysis in computational geometry [Su et al., 2015], optimal control such as swarm robotics and control systems [Chen et al., 2021, Krishnan and Martínez, 2018, Inoue et al., 2021], matching supply and demand networks [Lacombe et al., 2022], computer vision such as color transfer [Bai et al., 2023], and language translation [Xu et al., 2021].

Traditional methods for numerically computing the Wasserstein distance and geodesic require domain discretization that is often mesh-dependent (i.e., on regular grids or triangulated domains). Classical solvers such as Hungarian method [Kuhn, 1955], the auction algorithm [Bertsekas and Castanon, 1989], and transportation simplex [Luenberger and Ye, 2015], suffer from the curse-of-dimensionality and scale poorly for even moderately mesh-sized problems [Klatt et al., 2020, Genevay et al., 2016, Benamou and Brenier, 2000]. Entropic regularized OT [Cuturi, 2013] and the Sinkhorn algorithm [Sinkhorn, 1964] have been shown to efficiently approximate the OT solutions at low computational cost, handling high-dimensional distributions [Benamou et al., 2015]; however, high accuracy is computationally obstructed with a small regularization parameter [Altschuler et al., 2017, Dvurechensky et al., 2018]. Recently, machine learning methods to compute the Wasserstein geodesic for a *given* input pair of probability measures have been considered in [Liu et al., 2021, 2023, Pooladian et al., 2023, Tong et al., 2023], as well as *amortized* methods Lacombe et al. [2023], Amos et al. [2023] for generating static OT maps.

*Accepted for the 40$^{th}$ Conference on Uncertainty in Artificial Intelligence* (UAI 2024).

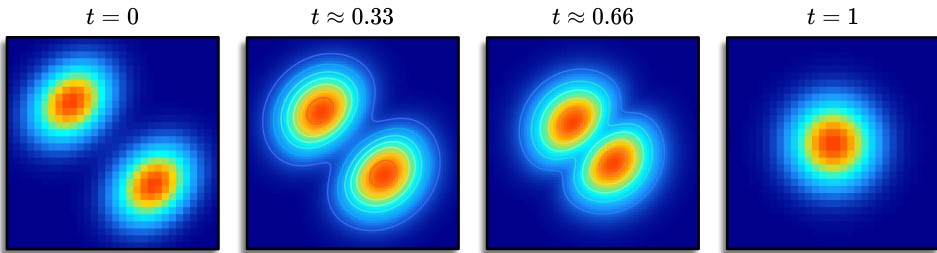

$$t = 0 \qquad t \approx 0.33 \qquad t \approx 0.66 \qquad t = 1$$

Figure 1: A geodesic at different spatial resolutions. Low-resolution inputs can be adapted into high-resolution geodesics (i.e., super-resolution) with our output mesh-invariant GeONet method.

A major challenge of using the OT-based techniques is that one needs to recompute the Wasserstein distance and geodesic for new input pair of probability measures. Thus, issues of scalability on large-scale datasets and suitability in the online learning setting are serious concerns for modern machine learning, computer graphics, and natural language processing tasks [Genevay et al., 2016, Solomon et al., 2015, Kusner et al., 2015]. This motivates us to tackle the problem of learning the Wasserstein geodesic from an *operator learning* perspective.

There is a recent line of work on learning neural operators for solving general differential equations or discovering equations from data, including DeepONet [Lu et al., 2021], Fourier Neural Operators [Li et al., 2020b], and physics-informed neural networks/operators (PINNs/PINOs) [Raissi et al., 2019, Li et al., 2021]. Those methods are mesh-independent, data-driven, and designed to accommodate specific physical laws governed by certain partial differential equations (PDEs).

**Our contributions.** In this paper, we propose a deep neural operator learning framework *GeONet* for the Wasserstein geodesic. Our method is based on learning the optimality conditions in the dynamic formulation of the OT problem, which is characterized by a coupled PDE system in the primal and dual spaces. Our main idea is to recast the learning problem of the Wasserstein geodesic from training data into an operator learning problem for the solution of the PDEs corresponding to the primal and dual OT dynamics. Our method can learn the highly non-linear Wasserstein geodesic operator from a wide collection of training distributions. GeONet is mesh-invariant, thus it is also suitable for zero-shot super-resolution applications on images, i.e., it is trained on lower resolution and predicts at higher resolution without seeing any higher resolution data [Shocher et al., 2018]. See Figure 1 for an example of a higher-resolution Wasserstein geodesic connecting two lower-resolution Gaussian mixture distributions.

Surprisingly, the training of our GeONet does not require the true geodesic data for connecting the two endpoint distributions. Instead, it only requires the training data as boundary pairs of initial and terminal distributions. The reason that GeONet needs much less input data is because its training process is implicitly informed by the OT dynamics such that the continuity equation in the primal space and Hamilton-Jacobi equation in the dual space must be simultaneously satisfied to ensure zero duality gap. Since the geodesic data are typically difficult to obtain without resorting to some traditional numerical solvers, the *amortized inference* nature of GeONet, where inference on related training pairs can be reused [Gershman and Goodman, 2014], has substantial computational advantage over standard computational OT methods and machine learning methods for computing the geodesic designed for single input pair of distributions [Peyré and Cuturi, 2019, Liu et al., 2021].

Once GeONet training is complete, the inference stage for predicting the geodesic connecting new initial and terminal data distributions requires only a forward pass of the network, and thus it can be performed in real-time. In contrast, standard OT methods re-compute the Wasserstein distance and geodesic for each new input distribution pair. This is an appealing feature of amortized inference to use a pre-trained GeONet for fast geodesic computation or fine-tuning on a large number of future data distributions. A detailed comparison between our proposed method GeONet with other existing neural operators and networks for learning dynamics from data can be found in Table 1.

## 2 BACKGROUND

### 2.1 OPTIMAL TRANSPORT PROBLEM: STATIC AND DYNAMIC FORMULATIONS

The optimal mass transportation problem, first considered by the French engineer Gaspard Monge, is to find an optimal map $T^*$ for transporting a source distribution $\mu_0$ to a target distribution $\mu_1$ that minimizes some cost function $c : \mathbb{R}^d \times \mathbb{R}^d \to \mathbb{R}$:

$$\min_{T:\mathbb{R}^d \to \mathbb{R}^d} \left\{ \int_{\mathbb{R}^d} c(x, T(x)) \, \mathrm{d}\mu_0(x) : T_\sharp \mu_0 = \mu_1 \right\}, \quad (1)$$

where $T_\sharp \mu$ denotes the pushforward measure defined by $(T_\sharp \mu)(B) = \mu(T^{-1}(B))$ for measurable subset $B \subset \mathbb{R}^d$. In this paper, we focus on the quadratic cost $c(x, y) = \|x - y\|_2^2$. The Monge problem (1) induces a metric, known as the

Table 1: We compare our method GeONet with other methodologies, including traditional neural operators, physics-based neural networks (PINNs) for learning dynamics, and traditional optimal transport solvers.

| Method characteristic | Neural operator w/o physics-informed learning | PINNs | Traditional OT solvers | GeONet (Ours) |
|---|---|---|---|---|
| operator learning | ✓ | | | ✓ |
| satisfies the associated PDEs | ✓ | ✓ | | ✓ |
| does not require known geodesic data | | ✓ | ✓ | ✓ |
| output mesh independence | ✓ | ✓ | | ✓ |

*Wasserstein distance*, on the space $\mathcal{P}_2(\mathbb{R}^d)$ of probability measures on $\mathbb{R}^d$ with finite second moments. In particular, the 2-Wasserstein distance can be expressed in the relaxed Kantorovich form:

$$W_2^2(\mu_0, \mu_1) := \min_{\gamma \in \Gamma(\mu_0, \mu_1)} \left\{ \int_{\mathbb{R}^d \times \mathbb{R}^d} \|x - y\|_2^2 \, \mathrm{d}\gamma(x, y) \right\}, \tag{2}$$

where minimization over $\gamma$ runs over all possible couplings $\Gamma(\mu_0, \mu_1)$ with marginal distributions $\mu_0$ and $\mu_1$. Problem (2) has the dual form (cf. Villani [2003]):

$$W_2^2(\mu_0, \mu_1) = \sup_{\varphi \in L^1(\mu_0), \, \psi \in L^1(\mu_1)} \left\{ \int_{\mathbb{R}^d} \varphi \, \mathrm{d}\mu_0 \right. \\ \left. + \int_{\mathbb{R}^d} \psi \, \mathrm{d}\mu_1 : \varphi(x) + \psi(y) \leqslant \|x - y\|_2^2 \right\}. \tag{3}$$

Problems (1) and (2) are both referred to as the *static OT* problems, which have a close connection to fluid dynamics. Specifically, the Benamou-Brenier dynamic formulation [Benamou and Brenier, 2000] expresses the Wasserstein distance as a minimal kinetic energy flow problem:

$$\frac{1}{2} W_2^2(\mu_0, \mu_1) = \min_{(\mu, \mathbf{v})} \int_0^1 \int_{\mathbb{R}^d} \frac{1}{2} \|\mathbf{v}(x, t)\|_2^2 \, \mu(x, t) \, \mathrm{d}x \, \mathrm{d}t$$

subject to $\partial_t \mu + \mathrm{div}(\mu \mathbf{v}) = 0, \mu(\cdot, 0) = \mu_0, \mu(\cdot, 1) = \mu_1$, (4)

where $\mu_t := \mu(\cdot, t)$ is the probability density flow at time $t$ satisfying the continuity equation (CE) constraint $\partial_t \mu + \mathrm{div}(\mu \mathbf{v}) = 0$ that ensures the conservation of unit mass along the flow $\{\mu_t\}_{t \in [0,1]}$. To solve (4), we apply the Lagrange multiplier method to find the saddle point in the primal and dual variables. In particular, for any flow $\mu_t$ starting from $\mu_0$ and terminating at $\mu_1$, the Lagrangian function for (4) can be written as

$$\mathcal{L}(\mu, \mathbf{v}, u) = \int_0^1 \int_{\mathbb{R}^d} \left[ \frac{1}{2} \|\mathbf{v}\|_2^2 \mu + (\partial_t \mu + \mathrm{div}(\mu \mathbf{v})) \, u \right] \, \mathrm{d}x \, \mathrm{d}t, \tag{5}$$

where $u := u(x, t)$ is the dual variable for CE. Using integration-by-parts under suitable decay conditions for $\|x\|_2 \to \infty$, we find that the optimal dual variable $u^*$ for the dynamic OT problem satisfies the Hamilton-Jacobi (HJ) equation

$$\partial_t u + \frac{1}{2} \|\nabla u\|_2^2 = 0, \tag{6}$$

and the optimal velocity vector field is given by $\mathbf{v}^*(x, t) = \nabla u^*(x, t)$. Hence, we obtained that the Karush–Kuhn–Tucker (KKT) optimality conditions for (4) are solution $(\mu^*, u^*)$ to the following system of PDEs:

$$\begin{cases} \partial_t \mu + \mathrm{div}(\mu \nabla u) = 0, \ \partial_t u + \frac{1}{2} \|\nabla u\|_2^2 = 0, \\ \mu(\cdot, 0) = \mu_0, \ \mu(\cdot, 1) = \mu_1. \end{cases} \tag{7}$$

In addition, if $\psi^*$ and $\varphi^*$ are the optimal Kantorovich potentials for solving the static dual OT problem (3), then the solution to the HJ equation (6) can be viewed as an interpolation $u(x, t)$ of the Kantorovich potentials between the initial and terminal distributions in the sense that $u^*(x, 1) = \psi^*(x)$ and $u^*(x, 0) = -\varphi^*(x)$ (both up to some additive constants). A detailed derivation of the primal-dual optimality conditions for the dynamical OT formulation is provided in Appendix B.

## 2.2 LEARNING NEURAL OPERATORS

Physics-informed neural networks (PINNs) [Raissi et al., 2019] aim to learn the solution of a PDE from data for a *given* input function $a$:

$$\partial_t u + \mathcal{D}_a[u] = 0 \tag{8}$$

subject to some boundary data $u_0$ and $u_T$, where $\mathcal{D}_a$ denotes a differential operator in space that may depend on the input function $a \in \mathcal{A}$. Different from the classical neural network learning paradigm that is purely data-driven, a PINN has less input data (i.e., some randomly sampled data points from the solution $u$ and the boundary conditions) since the solution operator $\Gamma^\dagger : \mathcal{A} \to \mathcal{U}$ is learned by obeying the induced physical laws governed by (8), and not from observations. Even though the PINN is mesh-independent, it only learns the solution for a *single* instance of the input function $a$ in the PDE (8). In order to learn the behavior of the inverse problem $\Gamma^\dagger : \mathcal{A} \to \mathcal{U}$ for an entire family of $\mathcal{A}$, we consider the operator learning perspective.

A neural operator generalizes a neural network that learns the mapping $\Gamma^\dagger : \mathcal{A} \to \mathcal{U}$ between infinite-dimensional function spaces $\mathcal{A}$ and $\mathcal{U}$ [Kovachki et al., 2021, Li et al., 2020a]. A notable example of operating learning is that $\mathcal{A}$ and $\mathcal{U}$ contain functions defined over a space-time domain $\Omega \times [0, T]$ with $\Omega \subset \mathbb{R}^d$, and the mapping of interest $\Gamma^\dagger$ is implicitly defined through a differential operator.

The idea of using neural networks to approximate any nonlinear continuous operator stems from the universal approximation theorem for operators [Chen and Chen, 1995, Lu et al., 2021]. In particular, we construct a parametric map by a neural network $\Gamma_\theta := \Gamma(\cdot; \theta) : \mathcal{A} \to \mathcal{U}$ for a finite-dimensional parameter $\theta \in \Theta$ to approximate the true solution operator $\Gamma^\dagger$. In this paper, we adopt the *DeepONet* architecture [Lu et al., 2021], which is suitable for their ability to learn mappings from pairings of initial input data to model $\Gamma^\dagger$. In the next subsection, we briefly discuss some basics of DeepONet architecture for modeling $\Gamma^\dagger$ and its enhanced version. Then, the neural operator learning problem is to find an optimal $\theta^* \in \Theta$ as a minimizer of the classical risk minimization problem

$$
\begin{aligned}
\min_{\theta \in \Theta} \mathbb{E}_{(a, u_0, u_T) \sim \nu} \Big[ & \big\| (\partial_t + \mathcal{D}) \Gamma_\theta(a) \big\|_{L^2(\Omega \times (0,T))}^2 \\
& + \lambda_0 \big\| \Gamma_\theta(a)(\cdot, 0) - u_0 \big\|_{L^2(\Omega)}^2 \\
& + \lambda_T \big\| \Gamma_\theta(a)(\cdot, T) - u_T \big\|_{L^2(\Omega)}^2 \Big],
\end{aligned}
\tag{9}
$$

where the input data $(a, u_0, u_T)$ are sampled from some joint distribution $\nu$. In (9), we minimize the PDE residual loss corresponding to $\partial_t u + \mathcal{D}_a[u] = 0$ while constraining the network by imposing boundary conditions. The loss function has weights $\lambda_0, \lambda_T > 0$. Given a finite set of samples $\{(a^{(i)}, u_0^{(i)}, u_T^{(i)})\}_{i=1}^n$, and data points randomly sampled in the space-time domain $\Omega \times (0, T)$, we may minimize the empirical loss analog of (9) by replacing $\|\cdot\|_{L^2(\Omega \times (0,T))}$ with the discrete $L^2$ norm over domain $\Omega \times (0, T)$. Computation of the exact differential operators $\partial_t$ and $\mathcal{D}_a$ can be conveniently exploited via automatic differentiation in standard deep learning packages.

## 2.3 DEEP OPERATOR NETWORKS

The DeepONet architecture [Lu et al., 2021] is based on the universal approximation theorem for operators [Chen and Chen, 1995], which says a general nonlinear continuous operator $\Gamma^\dagger$ may be approximated as follows:

$$
\Gamma^\dagger(u)(x, t) \approx \sum_{k=1}^p \mathcal{B}_k\big(u(x_1), \ldots, u(x_m); \theta\big) \cdot \mathcal{T}_k(x, t; \xi),
\tag{10}
$$

where $\mathcal{B}_k, \mathcal{T}_k$ are scalar elements of output of neural networks $\mathcal{B}, \mathcal{T}$, and $p$ is the number of such elements. For instance, we may take $\mathcal{B}$ and $\mathcal{T}$ as artificial neural networks parameterized by $\theta, \xi$ respectively. Networks $\mathcal{B}, \mathcal{T}$ are referred to as the *branch* and *trunk* networks, respectively.

The unstacked DeepONet in (10) is restricted to one input function $u$. In our problem, since we have two initial and terminal conditions, we consider an enhanced version of DeepONet [Tan and Chen, 2022], where the operator $\Gamma^\dagger$ is approximated using two branch networks to encode for input $u_0$ and $u_1$,

$$
\begin{aligned}
\Gamma^\dagger(u_0, u_1)(x, t) \approx & \sum_{k=1}^p \mathcal{B}_k^0\big(u_0(x_1), \ldots, u_0(x_m); \theta^0\big) \\
& \times \mathcal{B}_k^1\big(u_1(x_1), \ldots, u_1(x_m); \theta^1\big) \times \mathcal{T}_k(x, t; \xi).
\end{aligned}
\tag{11}
$$

In (11), the operator $\Gamma^\dagger$ is applied at the functions $u_0$ and $u_1$, and then evaluated at distinct locations $x_1, \ldots, x_m$ for the branch input.

## 3 OUR METHOD

We present *GeONet*, a geodesic operator network for learning the 2-Wasserstein geodesic $\{\mu_t\}_{t \in [0,1]}$ connecting $\mu_0$ to $\mu_1$. Let $\Omega \subset \mathbb{R}^d$ be the spatial domain where the probability measures are supported. For absolutely continuous probability measures $\mu_0, \mu_1 \in \mathcal{P}_2(\Omega)$, it is well-known that the constant-speed geodesic $\{\mu_t\}_{t \in [0,1]}$ between $\mu_0$ and $\mu_1$ is an absolutely continuous curve in the metric space $(\mathcal{P}_2(\Omega), W_2)$, which we denote as $\mathrm{AC}(\mathcal{P}_2(\Omega))$. Moreover, the geodesic $\mu_t$ solves the kinetic energy minimization problem in (4) [Santambrogio, 2015]. Some basic facts on the metric geometry structure of the Wasserstein geodesic and its relation to the fluid dynamic formulation are reviewed and discussed in Appendix C. In this work, our goal is to learn the non-linear operator

$$
\Gamma^\dagger : \mathcal{P}_2(\Omega) \times \mathcal{P}_2(\Omega) \to \mathrm{AC}(\mathcal{P}_2(\Omega)),
\tag{12}
$$

$$
(\mu_0, \mu_1) \mapsto \{\mu_t\}_{t \in [0,1]},
\tag{13}
$$

based on a training dataset $\{(\mu_0^{(1)}, \mu_1^{(1)}), \ldots, (\mu_0^{(n)}, \mu_1^{(n)})\}$. The core idea of GeONet is to learn the KKT optimality condition (7) for the Benamou-Brenier problem. Since (7) is derived to ensure the zero duality gap between the primal and dual dynamic OT problems, solving the Wasserstein geodesic requires us to introduce two sets of neural networks that train the coupled PDEs simultaneously. Specifically, we model the operator learning problem as an enhanced version of the unstacked DeepONet architecture [Lu et al., 2021, Tan and Chen, 2022] by jointly training three primal networks in (14) and three dual networks in (15) as follows:

$$
\begin{aligned}
\mathcal{C}(\mu_0, \mu_1)(x, t; \phi) = & \sum_{k=1}^p \mathcal{B}_k^{0, \mathrm{cty}}(\mu_0; \theta^{0, \mathrm{cty}}) \\
& \times \mathcal{B}_k^{1, \mathrm{cty}}(\mu_1; \theta^{1, \mathrm{cty}}) \times \mathcal{T}_k^{\mathrm{cty}}(x, t; \xi^{\mathrm{cty}})
\end{aligned}
\tag{14}
$$

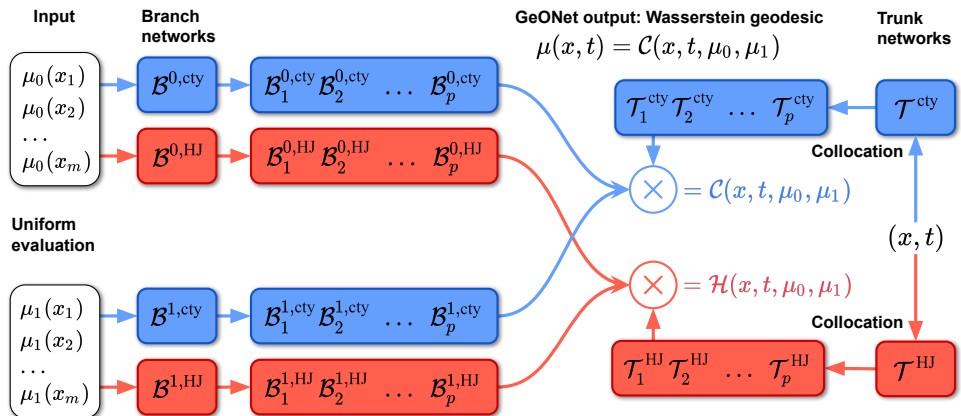

Figure 2: Architecture of GeONet. The solution to CE yields the geodesic. GeONet branches and trunks output vectors of dimension $p$, in which we perform multiplication among neural network elements to produce the solutions to CE and HJ.

and

$$\mathcal{H}(\mu_0, \mu_1)(x, t; \psi) = \sum_{k=1}^{p} \mathcal{B}_k^{0,\text{HJ}}(\mu_0; \theta^{0,\text{HJ}}) \quad (15)$$
$$\times \mathcal{B}_k^{1,\text{HJ}}(\mu_1; \theta^{1,\text{HJ}}) \times \mathcal{T}_k^{\text{HJ}}(x, t; \xi^{\text{HJ}}),$$

where $\mathcal{B}^{j,\text{cty}}(\mu_j(x_1), \ldots, \mu_j(x_m); \theta^{j,\text{cty}}) : \mathbb{R}^m \to \mathbb{R}^p$ and $\mathcal{B}^{j,\text{HJ}}(\mu_j(x_1), \ldots, \mu_j(x_m); \theta^{j,\text{HJ}}) : \mathbb{R}^m \to \mathbb{R}^p$ are *branch* neural networks taking $m$-discretized input of initial and terminal density values at $j = 0$ and $j = 1$ respectively, and $\mathcal{T}^{\text{cty}}(x, t; \xi^{\text{cty}}) : \mathbb{R}^d \times [0,1] \to \mathbb{R}^p$ and $\mathcal{T}^{\text{HJ}}(x, t; \xi^{\text{HJ}}) : \mathbb{R}^d \times [0,1] \to \mathbb{R}^p$ are *trunk* neural networks taking spatial and temporal inputs. Here $\Theta$ and $\Xi$ are finite-dimensional parameter spaces, and $p$ is the output dimension of the branch and truck networks. Denote parameter concatenations $\phi := (\theta^{0,\text{cty}}, \theta^{1,\text{cty}}, \xi^{\text{cty}})$ and $\psi := (\theta^{0,\text{HJ}}, \theta^{1,\text{HJ}}, \xi^{\text{HJ}})$. Then the primal operator network $\mathcal{C}_\phi(x, t, \mu_0, \mu_1) := \mathcal{C}(\mu_0, \mu_1)(x, t; \phi)$ for $\phi \in \Theta \times \Theta \times \Xi$ acts as an approximate solution to the CE, hence the true geodesic $\mu_t(x) = \Gamma^\dagger(x, t, \mu_0(x), \mu_1(x))$, while the dual operator network $\mathcal{H}_\psi(x, t, \mu_0, \mu_1)$ for $\psi \in \Theta \times \Theta \times \Xi$ corresponds to that of the associated HJ equation. The overall architecture of GeONet is shown in Figure 2.

In our GeONet implementation, we adopt a modified multi-layer perceptron (MLP) architecture, which has been shown to have great ability in improving performance for physics-informed DeepONets [Wang et al., 2021b]. We shall elaborate on this architecture in Appendix F.1 and describe our empirical findings with this modified MLP for GeONet in section 4.1.

To train the GeONet defined in (14) and (15), we minimize

the empirical loss function corresponding to the system of primal-dual PDEs and boundary residuals in (7) over the parameter space $\Theta \times \Theta \times \Xi$:

$$\phi^*, \psi^* = \operatorname{argmin}_{\phi,\psi \in \Theta \times \Theta \times \Xi} \quad \mathcal{L}_{\text{cty}} + \mathcal{L}_{\text{HJ}} + \mathcal{L}_{\text{BC}}, \quad (16)$$

where $\mathcal{L}_{\text{cty}}$ is the loss component in which the CE is satisfied in (17) and $\mathcal{L}_{\text{HJ}}$ is the HJ loss component in (18), while boundary conditions are incorporated in the $\mathcal{L}_{\text{BC}}$ term in (19). Automatic differentiation of our GeONet involves differentiating the coupled DeepONet architecture (cf. Figure 2) to compute the physics-informed loss terms.

Our loss function involves weight parameters $\alpha_1, \alpha_2, \beta_0, \beta_1$ to impose the physics-informed loss strength. Our coefficient tuning in the loss function is motivated and follows the general strategy outlined in [Wang et al., 2021b], where coefficients are tuned by examining errors and altered in an iterative procedure in which error is minimized. Boundary conditions are enforced to a greater extent, as precision with these affects precision in the physics loss.

We now illustrate our training procedure. The physics training is done via a *collocation* procedure, following [Raissi et al., 2019]. We randomly sample $N$ pairs $(x, t)$ uniformly within $\Omega \times [0, 1]$, where the CE and HJ expectation terms (17) and (18) in the loss function are approximated via a discrete empirical average. For the boundary terms (19), we evaluate $x$ among fixed locations with $\Omega$, typically a hypercube mesh, since these are where known boundary data is given, in which the neural operator is subsequently formulated and evaluated.

$$\mathcal{L}_{\text{cty}} = \alpha_1 \mathbb{E}_{(\mu_0, \mu_1) \sim (\mathcal{P}_2(\Omega), \mathcal{P}_2(\Omega))} \left[ ||\frac{\partial}{\partial t} \mathcal{C}_\phi(x, t) + \operatorname{div}(\mathcal{C}_\phi(x, t) \nabla \mathcal{H}_\psi(x, t))||_{L^2(\Omega \times (0,1))}^2 \right], \quad (17)$$

$$\mathcal{L}_{\text{HJ}} = \alpha_2 \mathbb{E}_{(\mu_0, \mu_1) \sim (\mathcal{P}_2(\Omega), \mathcal{P}_2(\Omega))} \left[ ||\frac{\partial}{\partial t} \mathcal{H}_\psi(x, t) + \frac{1}{2} ||\nabla \mathcal{H}_\psi(x, t)||_2^2||_{L^2(\Omega \times (0,1))}^2 \right], \quad (18)$$

$$\mathcal{L}_{\text{BC}} = \beta_0 \mathbb{E}_{(\mu_0) \sim (\mathcal{P}_2(\Omega))} \left[ ||\mathcal{C}_\phi(x, 0) - \mu_0||_{L^2(\Omega)}^2 \right] + \beta_1 \mathbb{E}_{(\mu_1) \sim (\mathcal{P}_2(\Omega))} \left[ ||\mathcal{C}_\phi(x, 1) - \mu_1||_{L^2(\Omega)}^2 \right]. \quad (19)$$

---

**Algorithm 1** End-to-end training of GeONet

---

**Input:** data pairs $(\mu_0^{(1)}\mu_1^{(1)}), \ldots, (\mu_0^{(n)}, \mu_1^{(n)})$; batch size $N$; initialization of the neural network parameters $\phi, \psi \in \Theta \times \Theta \times \Xi$; weight parameters $\alpha_1, \alpha_2, \beta_0, \beta_1$; domain $\Omega$ and branch domain (mesh) $\tilde{\Omega}$.; denote $i \in \{1, \ldots, N\}$.

1: **while** $\mathcal{L}_{\text{total}}$ has not converged **do**
2:      Independently draw $N$ sample points from $(x_\Omega^i, t^i) \in U(\Omega) \times U(0,1)$, $N$ points from $x_{\tilde{\Omega}}^i \in U(\tilde{\Omega})$, and $N$ density
     pairs from $\{(\mu_0^{(\ell)}, \mu_1^{(\ell)})\}_{\ell=1}^n$, possibly repeating.
3:      Compute $\mathcal{R}_{\text{cty},i} = \partial_t \mathcal{C}_{\phi,i} + \text{div}(\mathcal{C}_{\phi,i} \nabla \mathcal{H}_{\psi,i})$ at $(x_\Omega^i, t^i)$.          ▷ `continuity residual`
4:      Compute $\mathcal{R}_{\text{HJ},i} = \partial_t \mathcal{H}_{\psi,i} + \frac{1}{2}\|\nabla \mathcal{H}_{\psi,i}\|_2^2$ at $(x_\Omega^i, t^i)$.          ▷ `HJ residual`
5:      Compute $B_{0,i} = \mathcal{C}_{\phi,0,i} - \mu_0^{(i)}(x_{\tilde{\Omega}}^i)$, $B_{1,i} = \mathcal{C}_{\phi,1,i} - \mu_1^{(i)}(x_{\tilde{\Omega}}^i)$.          ▷ `boundary residual`
6:      Compute

$$\mathcal{L}_{\text{cty}} = \frac{\alpha_1}{N} \sum_{i=1}^N \mathcal{R}_{\text{cty},i}^2, \quad \mathcal{L}_{\text{HJ}} = \frac{\alpha_2}{N} \sum_{i=1}^N \mathcal{R}_{\text{HJ},i}^2,$$

$$\mathcal{L}_{\text{BC}} = \frac{1}{N} \sum_{i=1}^N (\beta_0 B_{0,i}^2 + \beta_1 B_{1,i}^2),$$

7:      Compute $\mathcal{L}_{\text{total}}(\phi, \psi) = \mathcal{L}_{\text{cty}} + \mathcal{L}_{\text{HJ}} + \mathcal{L}_{\text{BC}}$.
8:      Minimize $\mathcal{L}_{\text{total}}(\phi, \psi)$ to update $\phi$ and $\psi$.          ▷ `minimize the loss function`
9: **end while**

---

**Entropic regularization.** Our GeONet is compatible with entropic regularization, which is related to the Schrödinger bridge problem and stochastic control [Chen et al., 2016]. In Appendix D, we propose the *entropic-regularized GeONet* (ER-GeONet), which learns a similar system of KKT conditions for the optimization as in (7). In the zero-noise limit as the entropic regularization parameter $\varepsilon \downarrow 0$, the solution of the optimal entropic interpolating flow converges to solution of the Benamou-Brenier problem (4) in the sense of the method of vanishing viscosity [Mikami, 2004, Evans, 2010]. On one hand, adding a small entropy term (Laplacian) ensures the unique viscosity solution for the regularized HJ equation is smooth and benefits training. On the other hand, similarly as in the static OT problem, adding Laplacian approximates the OT flow (i.e., the Wasserstein geodesic is not solved exactly).

## 4 NUMERIC EXPERIMENTS

In this section, we perform simulation studies and a real-data example to demonstrate GeONet. Our code is publicly available at: `https://github.com/agracyk2/GeONet`.

**Error metric.** We use the $L^1$ error $\int_\Omega |\mathcal{C} - \mu|dx$ as our error metric to assess the performance, where $\mu := \mu(x, t)$ is a reference geodesic as proxy of the true geodesic without entropic regularization. The $L^1$ error integral is estimated by evaluating a discrete Riemann sum along a mesh and the reference is computed using the POT Python library [Solomon et al., 2015, Flamary et al., 2021]. Since $\int_\Omega |\mu|dx = 1$ for all time points, the $L^1$ error is relative, thus a meaningful metric essentially corresponding to the percentage error be-

tween the neural operator geodesic and the reference. We also consider the $L^2$ and Wasserstein error metric for predicted Wasserstein geodesics (see Appendix I).

### 4.1 INPUT AS CONTINUOUS DENSITY: GAUSSIAN MIXTURE DISTRIBUTIONS

Since finite mixture distributions are powerful universal approximators for continuous probability density functions [Nguyen et al., 2020], we first deploy GeONet on Gaussian mixture distributions over domains of varying dimensions. We learn the Wasserstein geodesic mapping between two distributions of the form $\mu_j(x) = \sum_{i=1}^{k_j} \pi_i \mathcal{N}(x|u_i, \Sigma_i)$ subject to $\sum_{i=1}^{k_j} \pi_i = 1$, where $j \in \{0, 1\}$ corresponds to initial and terminal distributions $\mu_0, \mu_1$, and $k_j$ denotes the number of components in the mixture. Here $u_i$ and $\Sigma_i$ are the mean vectors and covariance matrices of individual Gaussian components respectively. Due to the space limit, we defer simulation setups, model training details, and error metrics to Appendices G, H and I, respectively.

We examine errors in regard to an identity geodesic (i.e., $\mu_0 = \mu_1$), a random test pairing, and an out-of-distribution (OOD) pairing. The mesh-invariant nature of the output of GeONet allows zero-shot super-resolution for adapting low-resolution data into high-resolution geodesics, which includes initial data at $t = 0, 1$. Traditional OT solvers and non-operator learning based methods have no ability to do this, as they are confined to the original mesh. Thus, we also include a random test pairing on higher resolution than training data. The result is reported in Table 2.

**Univariate Gaussians.** We choose spatial domain $x \in \Omega = [0, 10]$ discretized into a 100-point mesh. We gen-

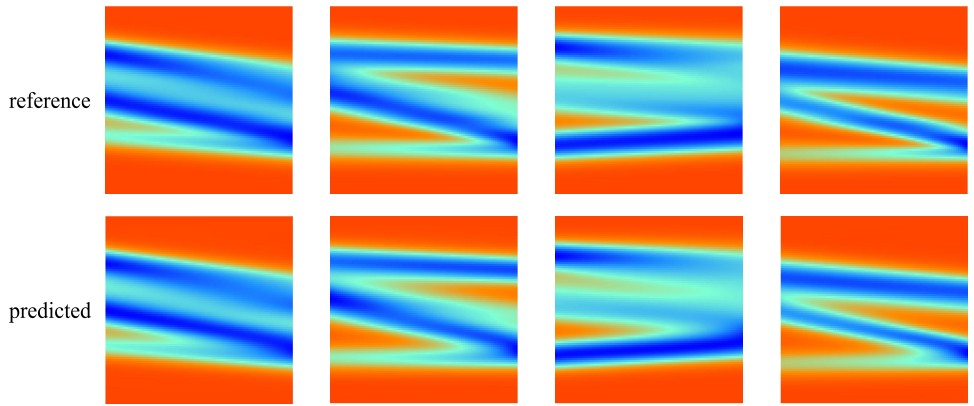

Figure 3: Four geodesics predicted by GeONet with reference geodesics computed by POT on test univariate Gaussian mixture distribution pairs with $k_0 = k_1 = 6$. The reference serves as a close approximation to the true geodesic. The vertical axis is space and the horizontal axis is time.

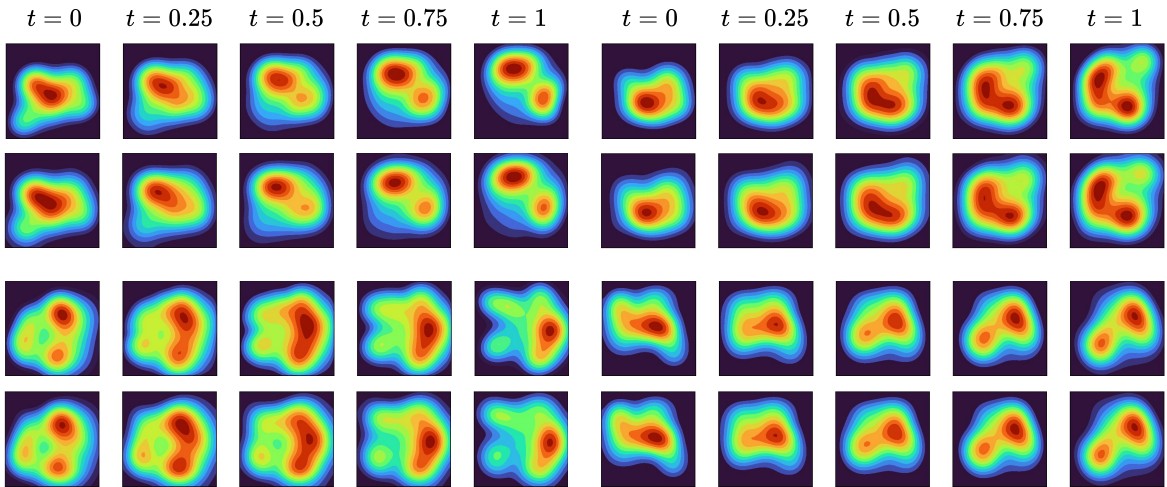

Figure 4: Geodesics predicted by GeONet on bivariate Gaussians over a square domain. The top of each pair is the reference solution computed by POT, and the bottom is GeONet.

erate $20,000$ training pairs $(\mu_0, \mu_1)$ of Gaussians, taking $k_j = 6$ for the number of Gaussians in each mixture. We take means $\mu_i \in [2,8]$ and variances $\Sigma_i \in [0.5, 0.6]$ uniformly. Empirically, we found a large batch size more suitable for training than a low one, so we take a batch size of $2,000$, meaning these many uniform collocation points are taken for both the PDE residuals and boundary points for each training iteration. We choose physical loss coefficient $\alpha_1 = 0.5, \alpha_2 = 0.25$, with boundary coefficients $\beta_0 = \beta_1 = 1$. We found these coefficients a good balance to enforce the physical constraint without sacrificing boundary restrictions after iterating these coefficients among $[0.05, 20]$ and examining the error. Additional training details are given in Appendix G.

**Bivariate Gaussians.** In our experiment, domain $\Omega = [0,5] \times [0,5] \subseteq \mathbb{R}^2$ was chosen, which was discretized into a $24 \times 24$ grid for GeONet input, meaning the branch

networks took vector input of $576$ in length for each in a non-convolutional architecture, but a convolutional architecture is also suitable in higher-dimensional cases as we see in Figure 9. We generate $5,000$ training pairs $(\mu_0, \mu_1)$. Recall that GeONet is mesh-invariant, so the $24 \times 24$ grids can be adapted to any higher resolution, which is used in Figure 4. We use a combination of low and high variance Gaussians in the mixture, 6 of which had variance in $[0.35, 0.4]$ and 6 in $[0.75, 0.9]$, giving a total of 12 Gaussians in each mixture in each pair. Covariances were in $[-0.1, 0.1]$. Additional training details are given in Appendix G.

**Training.** To compute the DeepONet derivatives, we take the inner product in the enhanced DeepOnet as in equations (14), (15), and subsequently use automatic differentiation after the inner products are taken. Alternatively, we experimented by computing a Hessian for the second-order derivatives, but this is costly in terms of memory, meaning

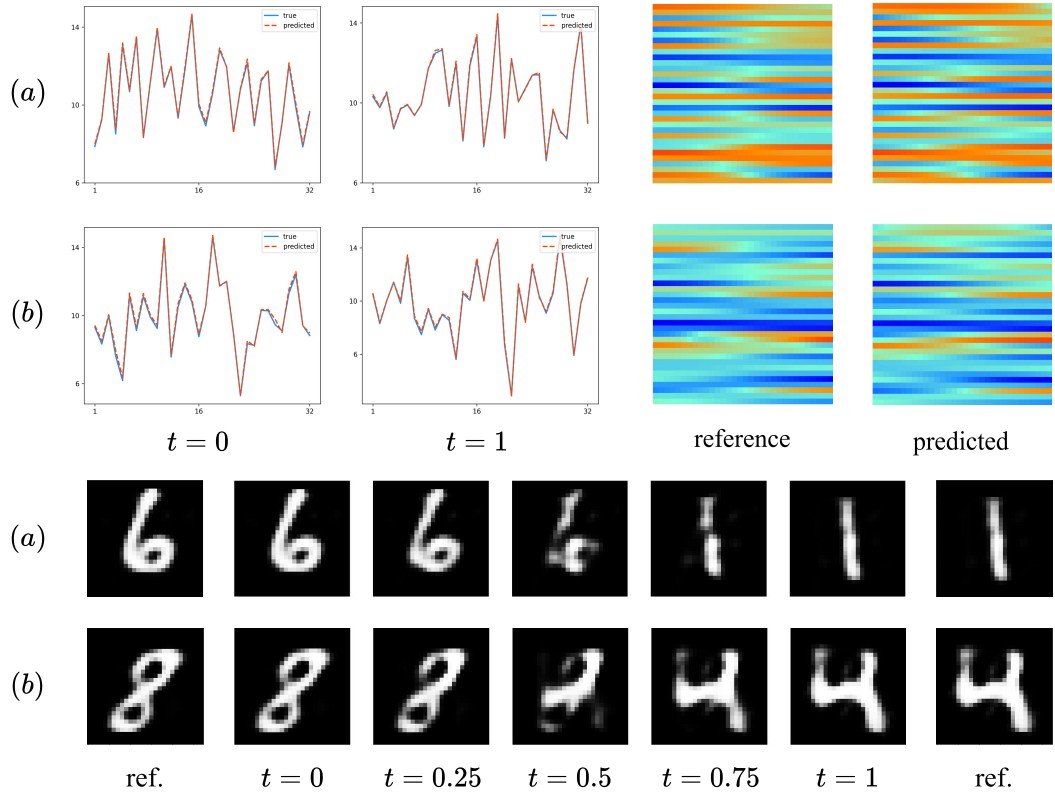

Figure 5: Beginning from the top left and going clockwise, we display the initial conditions in the encoded space, the geodesics in the encoded space, and the decoded geodesics as $28 \times 28$ images.

a large batch size cannot be used without a monumental memory cost, and so this method of differentiation is not viable.

We found that given sufficient data the GeONet with larger output dimensions slightly outperforms it with lower dimensions output. In the univariate Gaussian experiment, we take $p = 800$, which outperformed $p = 200$ by reducing training loss from approximately $2.5 \times 10^{-4}$ to $1.5 \times 10^{-4}$ and reducing test error by about $1\%$. In the bivariate experiment, changing $p = 400$ to $p = 800$ reduced training loss from approximately $2.1 \times 10^{-5}$ to $1.8 \times 10^{-5}$.

Architecture generally made some difference to training loss, but not significant, making a width of around 100-200 suitable for branches and trunks. For example, increasing branch width in the univariate experiment from 100 to 150 lowered training loss by approximately $4 \times 10^{-5}$. Increasing branch width to 200 and trunk width to 150 from 150 and 100 respectively had minimal effect, lowering training loss by about $1 \times 10^{-5}$. We found the modified MLP architecture preferable, lowering final training loss from approximately $3 \times 10^{-4}$ with standard architecture for univariate Gaussians.

## 4.2 INPUT AS POINT CLOUDS: GAUSSIAN MIXTURE DISTRIBUTIONS

GeONet can be applied to continuous densities made discrete. In scenarios with access to point clouds of data, we may use GeONet with discrete data made into empirical distributions. We test GeONet on an example of a Gaussian setup. We fix an initial and terminal distribution and sample discrete particles in $\Omega \subseteq \mathbb{R}^2$, as encompassed in [Liu et al., 2023]. The sampled particles are represented by empirical densities, in which we compare upon the transition of densities in the non-particle setting using POT as a baseline [Flamary et al., 2021]. The result is reported in Table 3 and an estimated geodesic example is shown in Figure 7. We observe that conditional flow matching (CFM) [Tong et al., 2023] and rectified flow (RF) [Liu et al., 2023] have 3-4 times comparably larger estimation errors than GeONet, except for the initial time $t = 0$, because this initial data is given and learned directly for RF and CFM. GeONet is the only framework among the comparison which captures the geodesic behavior to a considerable degree; however, we remark GeONet tends to smooth, or regularize, the solutions. Second, RF and CFM have the same fixed resolution as the input probability distribution pairing, while GeONet can estimate the density flows on higher resolution than the

Table 2: $L^1$ error of GeONet on 50 test data of univariate and bivariate Gaussian mixtures. We compute errors on cases of the identity geodesic, a random pairing in which $\mu_0 \neq \mu_1$, high-resolution random pairings refined to 200 and $75 \times 75$ resolutions in the 1D and 2D cases respectively, and out-of-distribution examples. We report the means and standard deviations as a percentage, making all values multiplied by $10^{-2}$ by those of the table.

| | GeONet $L^1$ error for Gaussian mixtures | | | | |
|---|---|---|---|---|---|
| **Experiment** | $t = 0$ | $t = 0.25$ | $t = 0.5$ | $t = 0.75$ | $t = 1$ |
| 1D identity | $2.67 \pm 0.750$ | $2.85 \pm 0.912$ | $3.04 \pm 1.02$ | $2.86 \pm 0.898$ | $2.63 \pm 0.696$ |
| 1D random | $4.92 \pm 2.00$ | $5.43 \pm 3.02$ | $5.76 \pm 3.56$ | $5.26 \pm 3.25$ | $4.65 \pm 1.50$ |
| 1D high-res. | $4.76 \pm 1.53$ | $5.49 \pm 3.00$ | $6.01 \pm 3.53$ | $5.59 \pm 2.99$ | $4.77 \pm 1.49$ |
| 1D OOD | $14.1 \pm 4.34$ | $18.8 \pm 5.96$ | $22.2 \pm 7.32$ | $19.2 \pm 6.14$ | $13.8 \pm 4.68$ |
| 2D identity | $6.50 \pm 1.15$ | $7.68 \pm 0.915$ | $7.69 \pm 0.924$ | $7.70 \pm 0.889$ | $6.42 \pm 1.11$ |
| 2D random | $6.59 \pm 1.01$ | $7.10 \pm 0.869$ | $7.13 \pm 0.892$ | $7.04 \pm 0.780$ | $6.33 \pm 0.835$ |
| 2D high-res. | $6.66 \pm 0.766$ | $7.71 \pm 1.26$ | $7.88 \pm 1.21$ | $7.59 \pm 0.979$ | $6.29 \pm 0.723$ |
| 2D OOD | $10.2 \pm 1.18$ | $9.82 \pm 1.12$ | $9.98 \pm 1.23$ | $9.67 \pm 1.03$ | $9.92 \pm 0.944$ |

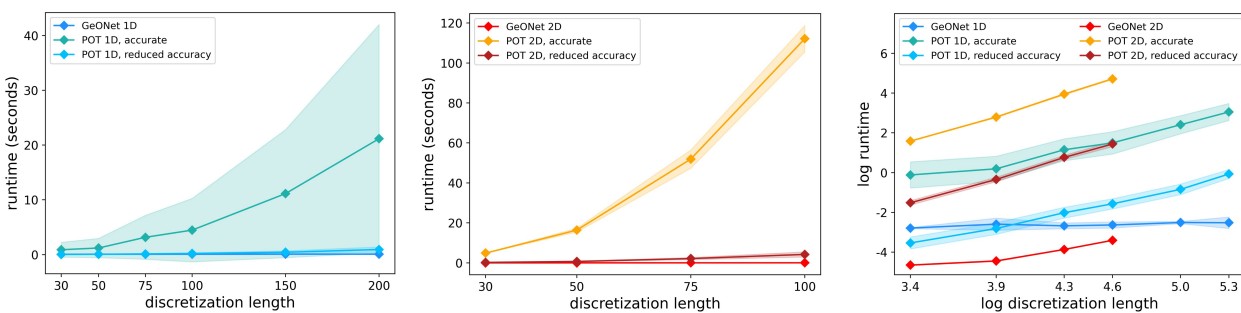

Figure 6: We compare GeONet to the classical POT library on 1D and 2D Gaussians in terms of mean and standard deviations of runtime on an unmodified scale as well as one that is log-log using discretization length in one dimension as the x-axis, taken over 30 pairs. We use 20-time steps for 1D and 5 for 2D. Finer meshes are omitted for 2D for computational reasonableness.

input pairing (cf. the third row in Figure 7).

### 4.3 A REAL DATA APPLICATION

Our next experiment was on the MNIST dataset of $28 \times 28$ images of single-digit numbers. It is difficult for GeONet to capture the geodesics between digits: MNIST resembles jump-discontinuous data, and is relatively piecewise constant otherwise, which is troublesome for physics-informed learning. To remedy our problems with MNIST, we use a pre-trained autoencoder to encode the MNIST digits into a low-dimensional representation $v \in \mathbb{R}^{32}$ with an encoder $\Phi$ and a decoder $\Phi^{-1} : v \to \mathbb{R}^{28} \times \mathbb{R}^{28}$ mapping the encoded representation into newly-formed digits resembling that which was fed into the encoder. The encoded data is made nonnegative via shifting upwards by a constant (we choose 10), and normalized over the domain to satisfy the density condition. This prepares the encoded data for GeONet input. We employ GeONet upon the encoded representations, learning the geodesic between highly irregular encoded data. The data can be decoded by unnormalizing and shifting

downwards by the arbitrary constant. For normalization constants at $t \neq 0, 1$, we use interpolation between the constants at $t = 0, 1$.

Table 6 reports the $L^1$ errors for geodesic estimated in the encoded space and recovered images in the ambient space. As expected, the ambient-space error is much larger than the encoded-space error, meaning that the geodesics in the encoded space and ambient image space do not coincide. Figure 5 shows the learned geodesics in the encoded space and decoded images on the geodesics.

### 4.4 RUNTIME COMPARISON

Our method is highlighted by the fact that it is almost instantaneous: it is highly suitable when many geodesics are needed quickly, or over fine meshes. Traditional optimal transport solvers are greatly encumbered when evaluated over a fine grid, but the mesh-invariant output nature of GeONet bypasses this. In Figure 6, we illustrate GeONet versus POT, a traditional OT library. GeONet greatly out-

Table 3: $L^1$ error between GeONet, the conditional flow matching (CFM) library's optimal transport solver Tong et al. [2023], and rectified flow (RF) Liu et al. [2023], using POT again as a baseline for comparison. All values are multiplied by $10^{-2}$ to those of the table.

| Experiment | $L^1$ comparison error on 2D Gaussian mixture point clouds | | | | |
| | $t = 0$ | $t = 0.25$ | $t = 0.5$ | $t = 0.75$ | $t = 1$ |
| --- | --- | --- | --- | --- | --- |
| GeONet | $22.9 \pm 1.08$ | $28.8 \pm 1.01$ | $30.0 \pm 1.10$ | $29.6 \pm 0.877$ | $22.6 \pm 1.02$ |
| CFM | $0.0 \pm 0.0$ | $94.1 \pm 3.68$ | $98.9 \pm 2.41$ | $91.8 \pm 4.15$ | $75.9 \pm 3.77$ |
| RF | $0.0 \pm 0.0$ | $103 \pm 2.48$ | $112 \pm 3.61$ | $112 \pm 5.03$ | $91.3 \pm 3.79$ |

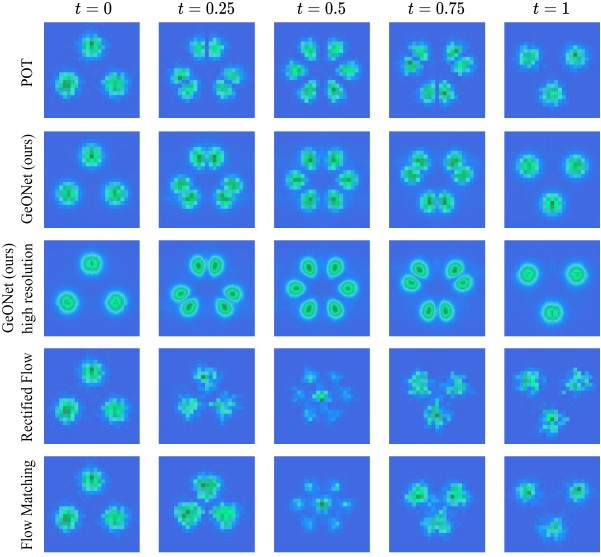

Figure 7: We compare to GeONet to the alternative methodology in a discrete setting, using POT as ground truth. GeONet is the only method among the comparison which captures the geodesic behavior among the translocation of points.

performs POT for fine grids in terms of runtime, especially if POT is used to compute an accurate solution. Even when POT is used with equivalent accuracy, GeONet still outperforms, most illustrated in the log-log plot. The log-log plot also demonstrates that our method speeds computation up to orders of magnitude. We restrict the accuracy of POT by employing a stopping threshold of $0.5$ for 1D and $10.0$ for 2D. We found these choices were comparable to GeONet, remarking a threshold of $10.0$ in the 2D case is sufficiently large so that even larger thresholds have limited effect on error.

### 4.5 OUT-OF-DISTRIBUTION GENERALIZATION

We discuss GeONet on out-of-distribution data in the test setting upon Gaussian mixture data. Our error results are provided in Table 2. For univariate Gaussians, we choose

means in $[1, 9]$, which was expanded from the domain $[2, 8]$ in training. This increased relative error by about $10\%$. Variances were in $[0.3, 0.4]$. A 100-point mesh is used for evaluations with POT regularization parameter $\epsilon = 6 \times 10^{-4}$. For 2D Gaussians, we test on 16 mixture components (training has 12). Means were in $[0.6, 4.4] \times [0.6, 4.4]$, which was expanded from $[0.8, 4.2] \times [0.8, 4.2]$ in training. There were 8 components in the mixture with variance in $[0.25, 0.3]$ and the other 8 in $[0.65, 0.8]$, which have lower variances than those in training. Covariances are within $[-0.15, 0.15]$ for off-diagonal components in each covariance matrix. Evaluations were over a $24 \times 24$ mesh, the same used as neural operator input.

### 4.6 LIMITATIONS

There are several limitations we would like to discuss. First, GeONet's branch network input exponentially increases in spatial dimension, necessitating extensive input data even in moderately high-dimensional scenarios. One strategy to mitigate this is through leveraging low-dimensional data representations as in the MNIST experiment. GeONet is near instantaneous for any dimension, but its dimension-based restrictions to perform are mostly hindered by the ability to handle neural network input in the branches. Second, GeONet mandates predetermined evaluation points for branch input, a requisite grounded in the pairing of initial conditions. It is of interest to extend GeONet to include training input data pairs on different resolutions. Third, given the regularity of the OT problem [Hütter and Rigollet, 2021, Caffarelli, 1996], developing a generalization error bound for assessing the predictive risk of GeONet is an important future work. Finally, the dynamical OT problem is closely connected to the mean-field planning with an extra interaction term [Fu et al., 2023]. Extending the current operator learning perspective to such problems would be interesting.

### Acknowledgements

Andrew Gracyk was supported by the NSF under grant No. 1922758. Xiaohui Chen was partially supported by NSF CAREER grant DMS-2347760, NSF grant DMS-2413404, and a gift from the Simons Foundation.

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
