# OpenReview forum: "GeONet: a neural operator for learning the Wasserstein geodesic"
_auai.org/UAI/2024/Conference — UAI 2024 poster_

### Official Review · Reviewer_HV3E · 2024-02-28

**Q2-1 Originality-Novelty:** 3
**Q2-2 Correctness-Technical Quality:** 3
**Q2-5 Clarity Of Writing:** 4

**Q1 Summary And Contributions:**

This paper presents GeONet, a method that learns an OT (Wasserstein geodesic) mapping from an initial to a terminal probability distribution. The problem is formulated as estimating the solution to a system of two coupled PDEs. Inspired by operator learning in neural PDE solving, the authors adopt the DeepONet architecture. This allows for a certain kind of "resolution-agnostic" model. The method is tested on synthetic mixture-of-Gaussian datasets as well as MNIST. A core advantage is a considerable reduction of inference time.

**Q2-3 Extent To Which Claims Are Supported By Evidence:**

3: Good: the main claims are supported by convincing evidence (in the form of adequate experimental evaluation, proofs, (pseudo-)code, references, assumptions).

**Q2-4 Reproducibility:**

3: Good: key resources (e.g. proofs, code, data) are available and key details (e.g. proofs, experimental setup) are sufficiently well-described for competent researchers to confidently reproduce the main results.

**Q3 Main Strengths:**

* The paper presents original ideas an a novel approach of attack to the OT problem.
* The paper is presented well, providing clear definitions and intuition about the methodology.
* The paper is honest about its limitations.

**Q4 Main Weakness:**

* Although OT theory has a core place in ML research, I am not sure in what scenarios the provided method (and experiments) can be applied. That is, I find the experiments somewhat synthetic.

**Q5 Detailed Comments To The Authors:**

* I would suggest the authors provide a bit of motivation to the work, in what settings can the provided method lead to new results or insights?
* Forgive my ignorance, as I'm not super familiar with the topic, but what part of the method ensures that, e.g. in Figure 4, the intermediate distribution around t=0.5 takes this shape, given that the endpoints seem to be identical?
* What exactly do the authors report with the $L_1$ error, how is it defined and computed?
* Is it realistic to assume that ground truth densities can be evaluated for loss computation in e.g. (20)?
* Will the source code be released?

**Q9 Complying With Reviewing Instructions:**

Yes

---

> ### Author Rebuttal · Authors · 2024-04-07
>
> Q5#1 [motivation] Learning the (continuous/dynamical) Wasserstein geodesic based on the OT theory is a practically important task in many science and engineering domains, including:
>
> (i) developmental trajectory reconstruction in cell reprogramming [1];
>
> (ii) 3D warping for shape analysis in computational geometry [2] (e.g., from an Amarillo to a unit sphere);
>
> (iii) optimal control [3,4] (e.g., swarm robotics and control systems);
>
> (iv) matching supply and demand networks [5];
>
> (v) computer vision [6] (e.g., color transfer and super-resolution);
>
> (vi) language translation [7] (e.g., matching word embeddings from English to French),
>
> among many others. We will accordingly revise the introduction in the next version.
>
> [1] https://doi.org/10.1016/j.cell.2019.01.006
>
> [2] https://doi.org/10.1109%2FTPAMI.2015.2408346
>
> [3] https://doi.org/10.23919/ACC50511.2021.9483194
>
> [4] https://doi.org/10.1109/CDC.2018.8619816
>
> [5] https://arxiv.org/abs/2102.12178
>
> [6] https://dcoeurjo.github.io/OTColorTransfer/
>
> [7] https://doi.org/10.18653/v1/2021.acl-long.571
>
>
> Q5#2 In Fig. 4, the endpoint distributions at t=0 and t=1 have different centroids (component mean vectors), while they have the same number of components and covariance matrix. The centroids at t=1 are a rotation version of those at t=0. So the two endpoints are different distributions, and the geodesic connecting t=0 and t=1 first splits the initial centroid and then merges into the terminal centroid locations.
>
>
> Q5#3 [L1 error definition & interpretation] The L1 error is defined as $\int_\Omega |C - \mu| dx$, where $\mu := \mu(x, t)$ is a reference geodesic (usually entropic regularized) as a proxy of the true geodesic and $C$ is the predicted geodesic from GeONet’s continuity equation network. In numeric experiments, the L1 error integral is estimated by evaluating a discrete Riemann sum along a mesh, and the reference is computed using the Convolutional Wasserstein Barycenter framework within the POT Python library [8,9]. Because $\int_\Omega \mu dx = 1$ for all $t \in [0, 1]$, the L1 error is relative, a meaningful metric representing the percentage error between GeONet and the reference geodesics.
>
> [8] https://doi.org/10.1145/2766963
>
> [9] https://pythonot.github.io/
>
>
> Q5#4 [true density] The ground truth density can be computed in simulation or estimated from real data. For example,
>
> (i) Gaussian mixture distributions (Sec. 4.1 & 4.2): we evaluated the training data probability density function for the two endpoints on a grid. Specifically (as listed in the Appendix), the grid domain for 1D case is [0,10]. The domain for 2D case is [0,5]x[0.5].
>
> (ii) MNIST data (Sec. 4.3): we used a pre-trained encoder to first embed the MNIST data into a feature of 32 dimensions. Then the features are shifted to a non-negative function and then normalized as a probability density function. After this, GeONet was applied to learn the geodesic in the latent space.
>
> (iii) CIFAR-10 (cf. https://drive.google.com/file/d/1f4wksoOux3TYM4xTisw9GNDm0yHYc29I/view?usp=drive_link): we normalized the image pixel values (for each R/G/B channel) to a total mass value of one. Thus each CIFAR-10 image represents a probability density on on its 32*32 domain. Then we learn the geodesic for each channel and recover the super-resolution version of the CIFAR-10.
>
>
> Q5#5 [code release] We have uploaded most source code in Supplementary Material. We will also release it on our GitHub website after the paper decision.

---

### Official Review · Reviewer_7EUS · 2024-03-17

**Q2-1 Originality-Novelty:** 3
**Q2-2 Correctness-Technical Quality:** 2
**Q2-5 Clarity Of Writing:** 3

**Q1 Summary And Contributions:**

The paper introduces GeONet, a deep neural operator framework designed to learn the Wasserstein geodesic, which is the most efficient path connecting initial and terminal distributions in the space of probability measures. Traditional computational methods for determining the Wasserstein distance and geodesic face challenges such as the curse of dimensionality and dependence on mesh-specific domain discretization. GeONet overcomes these issues by learning a mesh-invariant mapping from pairs of distributions to their connecting geodesic, based on the saddle point optimality conditions of the dynamic optimal transport (OT) problem characterized by a coupled Partial Differential Equation (PDE) system. This enables real-time inference and deployment in online learning settings, offering a significant computational advantage over standard OT solvers. GeONet's effectiveness is demonstrated through experiments with Gaussian mixture distributions and the MNIST dataset, showcasing its capability to accurately compute Wasserstein geodesics with reduced computational cost, making it a promising tool for applications in machine learning, computer graphics, and natural language processing where understanding the geometric relationships between complex data distributions is crucial.

**Q2-3 Extent To Which Claims Are Supported By Evidence:**

3: Good: the main claims are supported by convincing evidence (in the form of adequate experimental evaluation, proofs, (pseudo-)code, references, assumptions).

**Q2-4 Reproducibility:**

2: Fair: key resources (e.g. proofs, code, data) are unavailable but key details (e.g. proof sketches, experimental setup) are sufficiently well-described for an expert to confidently reproduce the main results.

**Q3 Main Strengths:**

1. The description of the background and problem setting is clear enough to help the reader understand the problem easily.
2. The description of the method is straightforward to read.
3. The experimental results demonstrate the superiority of the proposed method.
4. The proposed method is innovative.

**Q4 Main Weakness:**

1. The experimental analysis should be extended to include real datasets across various scientific problems to justify the model's capabilities and generalizability.

2. The experimental analysis should include more baselines to demonstrate the superiority of the proposed method.

**Q5 Detailed Comments To The Authors:**

Same as weakness

**Q9 Complying With Reviewing Instructions:**

Yes

---

> ### Author Rebuttal · Authors · 2024-04-07
>
> Thank you for the suggestions! In the rebuttal, we added one more real dataset (CIFAR-10) and several simulation studies (3D Gaussian mixture and more OOD setups).
>
> Q4#1 [CIFAR-10] We added an example of super-resolution on CIFAR-10 images by GeONet. Note that traditional OT solvers cannot do this task. CIFAR-10 is 32*32 colored images, and we directly apply GeONet on the 1024 pixels without embedding to a lower-dimensional latent space (like we did in MNIST). The result can be viewed via
>
> https://drive.google.com/file/d/1f4wksoOux3TYM4xTisw9GNDm0yHYc29I/view?usp=drive_link
>
> where each row represents the original and higher-resolution initial (t=0) and (t=1) images.
>
>
> Q4#2 [3D Gaussian mixtures] In the rebuttal, we added a new 3D Gaussian mixture simulation and extra OOD settings. For the extra OOD setups and their testing errors, please refer to our reply to *Reviewer grLw Q4*.
>
> For the 3D Gaussian mixture, to the best of our try, POT is not applicable to produce a geodesic on the 3D grid in hypercube [0,4]x[0,4]x[0,4]. Regardless, since boundary conditions (input data) are always known, we can still evaluate the L1 error of GeONet on boundaries t=0 and t=1. The L1 errors (multiplied by 10^{-2}) are reported as follows:
>
> t=0:         $34.190 \pm 10.638$
>
> t=1:         $35.395 \pm 11.966$
>
> Detailed 3D Gaussian mixture setups, as well as also the predicted geodesic for 3 testing pairs by GeONet, are described and shown in the linked file (https://drive.google.com/file/d/1f4wksoOux3TYM4xTisw9GNDm0yHYc29I/view?usp=drive_link).

---

### Official Review · Reviewer_yMMr · 2024-03-18

**Q2-1 Originality-Novelty:** 3
**Q2-2 Correctness-Technical Quality:** 2
**Q2-5 Clarity Of Writing:** 2

**Q1 Summary And Contributions:**

This study proposes a neural operator learning method for Wasserstein geodesic. To solve the OT problem, authors use Benamou-Brenier dynamic formulation that transforms original problem into PDE solving problem. And DeepONet architecture was employed to solve the PDE. 1D and 2D Gaussian mixture cases were showcased along with MNIST dataset.

**Q2-3 Extent To Which Claims Are Supported By Evidence:**

3: Good: the main claims are supported by convincing evidence (in the form of adequate experimental evaluation, proofs, (pseudo-)code, references, assumptions).

**Q2-4 Reproducibility:**

3: Good: key resources (e.g. proofs, code, data) are available and key details (e.g. proofs, experimental setup) are sufficiently well-described for competent researchers to confidently reproduce the main results.

**Q3 Main Strengths:**

This study is perhaps the first approach in neural operator learning for Wasserstein geodesic. If so, it can pave the way for subsequent work in this line of research that can improve performance.

**Q4 Main Weakness:**

1. The manuscript has a serious issue in the presentation: the important concept is not so well presented that it does not read well.
2. Sect. 1, A major challenge of using the OT-based techniques is that one needs to recompute the Wasserstein distance and geodesic for new input pair of probability measures. => The previous paragraph explains about ML methods for computing Wasserstein distance. These methods are train once, use many times approach. So you don't need to recompute all the time.
Similarly, 'computational advantage over standard computational OT methods and machine learning methods for computing the geodesic designed for single input pair of distributions' => I understand the computational advantage over standard OT, but the machine learning methods are fast and accurate enough once trained.
3. There are four regulation parameters (\alpha's and \beta's), which may make the training really challenging.
4. At the end of Sect. 3, Entropic regularization is explained. However, its pros and cons were not discussed at all. Does it have the same tradeoff as the entropic regularized OT mentioned in Sect. 1?
5. Table 2 is difficult to interpret. Why does the error exist in t=0 and t=1 when these are initial and terminal distributions? How was OOD constructed here?
6. Sect. 4.4, Even when POT is used to equivalent accuracy, GeONet still outperforms, most illustrated in the log-log plot.  => POT library contains Entropic regularized OT. I wonder why authors didn't try that approach for runtime comparison.
7. Sect. 4.6, While traditional geodesic solvers primarily handle one or two dimensions, => This is wrong. POT solvers can handle higher dimensions as well.
8. Sect. 4.6, GeONet offers a versatile alternative, accommodating any dimension at the cost of potential computational precision. => The paper showed its performance only up to 2D.
9. The proposed method seems to only work well for basic distributions such as mixture of Gaussians at 2D maximum. Realistic data needs the help of low-dimensional embedding.

**Q5 Detailed Comments To The Authors:**

Sect. 3 is written in a very confusing way. You are given with n pairs of training set and Eq. 18 through 20 all of a sudden has the denominator N. C and H both have the subscript i and they have L2 norm in 18 through 20. The training process is described almost at the end (One iterate of our training procedure is as follows...), but until then it's really difficult to follow.

Other parts of writing should be improved a lot as stated follows:

1. pg. 2, and PINOs -> / (PINOS)
2. Sect 2.1, c(x,y) = 1/2||x-y||_2^2 => 1/2 should be dropped because Eq. 2's cost term is simply ||x-y||_2^2
3. Eqs. (3) and (4) should be consolidated. The same for 13 and 14; 15 and 16
4. pg. 3, both referred as -> both referred to as
5. pg. 3 defines continuity equation as CE, just use CE after that throughout the paper
6. In Eq. 7, u should be replaced with u^*
7. pg. 3, KKT optimality conditions for (6) => Note that Eq. 6 itself doesn't have any constraints. So 'KKT conditions for (6)' sounds weird.
8. pg. 3 defines Hamilton-Jacobi as HJ, just use HJ after that throughout the paper
9. pg. 3 R-column mentions Kantorovich potential, but its definition appears two lines later. It'd be better if its definition immediately follows.
10. Sect. 2.2, implicitly defined through certain differential operator => I believe neural operator learning is more general than finding relationship from differential equations. So restricting it to 'certain differential operator' doesn't sound correct.
11. Eq. 9, PINN itself is not a neural operator learning, which can confuse readers. I suggest presenting this example (Eq. 9) as a motivation first and expand it to a neural operator learning.
12. D := D(a) -> D[a] / Note that you don't need to define D here, as it's never used afterward.
13. has to obeys -> has to obey
14. \Gamma: A x \Theta -> U doesn't sound right. \Theta is a simply parameter in NN you learn by training and you search for optimal \thata*. As such, \Gamma cannot take \Theta as its domain. I suggest that in Eq. 10, you represent the parameter as \Gamma(a; \theta) instead of \Gamma(a, \theta). The same is suggested for subsequent equations such as (13), (15), etc.
15. pg. 4, a finite sample -> a finite set of samples
16. Sect. 3, connecting \mu_0 \mu_1 from the distance W_2 => Learning the geodesic from the W_2 distance doesn't make sense. In fact, you can compute the distance after you find the geodesic.
17. Sect. 3, hence the true geodesic \Gamma^\dagger( ) :=  => \Gamma^\dagger is a mapping that takes function space A. So this presentation looks incorrect. Plus, why do you define \Gamma^\dagger( ) with := notation again?
18. pg. 5, and C_\phi, t, i( ) definition is redundant and should be deleted.
19. Fig. 2 caption, continuity solution yields the geodesic -> the solution to CE
20. Sect. 3, Spatial-temporal -> Spatio-temporal
21. Sect 4.2, an an example -> an example
22. POT as a baseline => Provide citation for POT
23. Sect. 4.3, institute GeONet upon => I don't understand what this means
24. Fig. 5 caption, (a) and (b) correspond => There's no (a) and (b)
25. Fig. 6 caption, runtime on both an -> runtime on an
26. Appendix, G SPECIALIZED AERCHITECTURES -> G SPECIALIZED ARCHITECTURES

**Q9 Complying With Reviewing Instructions:**

Yes

---

> ### Author Rebuttal · Authors · 2024-04-07
>
> Q4#1 & Q5 [presentation & detailed comments] We apologize for the writing in Sec. 3. We will restructure the presentation in the next version. We will also collectively correct the typos and revise according to your suggestions.
>
> Q4#2 [existing literature] Existing ML methods in literature can be categorized into:
>
> 1. Amortized: learning the Wasserstein barycenter [1] and static OT map [2].
>
> 2. Non-amortized: learning Wasserstein geodesic between a *given* pair of probability distributions [3].
>
> Our work is the first to propose an amortized method based on operator learning for geodesic connecting new unseen / testing pairs of distributions.
>
> [1] arXiv:2102.12178
>
> [2] arXiv:2206.05262
>
> [3] arXiv:2102.02992
>
> Q4#3 [tuning] Our coefficient tuning in the loss function is motivated and follows the general strategy in [5]: boundary conditions are enforced to a greater extent, as precision with these also affects precision in the physics loss. Our coefficients are experimented upon so that loss is empirically satisfied under selected certain values, as done in [5]. Boundary loss coefficients are taken equal or greater than physics coefficients, which demonstrates greater success for us as well as in other training instances, as demonstrated on page 13 of [5] and page 32 of [4].
>
> [4] arXiv:2308.08468
>
> [5] arXiv:2103.10974
>
> Q4#4 [entropic regularization]
> Pro: adding a small entropy term (Laplacian) ensures the unique viscosity solution for the dual regularized HJ equation is smooth and benefits training. (Note that the HJ equation is first-order and its solution is non-smooth in general.)
> Con: as in the static OT problem, adding Laplacian approximates the OT flow (not solving the Wasserstein geodesic exactly).
>
>
> Q4#5 [boundary loss] Boundary conditions are not strictly enforced (as equality constraints). Rather, they are lifted to the loss function and GeONet minimizes the total loss (boundary + PDE) as an unconstrained optimization problem that can be tackled by standard deep neural networks.
>
> [OOD construction] Training data generation setups for 1D and 2D Gaussian mixtures are described in Appendix I in the Supplementary Material. Here, we summarize as:
> (i) 1D: 6 mixture components(in both endpoint distributions),
>  means uniformly sampled from [2, 8], and variance from [0.5, 0.6].
> (ii) 2D: 12 mixture components, means from [0.8,4.2]x[0.8,4.2], and 6 variances from [0.35, 0.4] and 6 variances from [0.75, 0.9]. All covariances were chosen from [-0.15,0.15] for each off-diagonal component.
>
> For OOD construction, we have correspondingly the following setups:
> (i) 1D: variance shift to [0.3, 0.4], with means in [1,9], a slightly expanded interval.
> (ii) 2D: (diagonal) variance shift from [0.35, 0.4] to [0.25, 0.3] for 6 components and from [0.75, 0.9] to [0.65, 0.8] for the other 6 components.
>
> Q4#6 [POT] Our baseline is indeed the POT solver with the entropy-regularized OT cost and a small regularization parameter eps:
>
> 0.0006 for 1D mixtures;
>
> 0.004 for 2D mixtures;
>
> 0.0003 for encoded MNIST;
>
> 0.002 for MNIST in ambient space.
>
> For this reason, we called the geodesic computed by POT the *reference* geodesic (rather than the true unregularized one). So the runtime comparison between our GeONet (without regularization) and POT (with entropy) is fairly reasonable.
>
> Q4#7 [dimensionality] We apologize for the confusion. Here, the dimension refers to the domain dimension D. In theory, POT can compute higher dimensional OT maps on grids (e.g., image pixels) or point clouds. However, it suffers from curse-of-dimensionality (for a vanishing regularization parameter). For example, the grid size scales as m^D to cover the domain, where m is the mesh size in 1-dim. For large D and moderate resolution m, the problem scale for traditional OT solvers (including POT) has an exponential increase in dimension D (i.e., curse-of-dimensionality), and thus their computational cost is prohibitive.  In contrast, GeONet is mesh-invariant since it is an (infinity-dim) operator learning approach. So its computational cost doesn’t scale as the traditional OT solvers. In the rebuttal, we added a 3D example for GeONet (https://drive.google.com/file/d/1f4wksoOux3TYM4xTisw9GNDm0yHYc29I/view?usp=drive_link), where POT cannot practically compute the geodesic. We apologize for this confusion and will clarify in the next version.
>
> Q4#8 [3D Gaussian mixtures] *Review 7EUS (Q4#2 [3D Gaussian mixtures])* raised a similar question on adding more baseline examples to demonstrate GeONet superiority. We performed a new 3D Gaussian mixture simulation and please find details therein.
>
> Q4#9 3D [CIFAR-10, no embedding] *Reviewer 7EUS (Q4#1 [CIFAR-10])* raised a similar concern on extra real dataset. We added a CIFAR-10 experiment on the pixel level. Please refer details therein.

---

### Official Review · Reviewer_grLw · 2024-03-23

**Q2-1 Originality-Novelty:** 3
**Q2-2 Correctness-Technical Quality:** 3
**Q2-5 Clarity Of Writing:** 3

**Q1 Summary And Contributions:**

This paper proposes GeONet which is a deep neural operator learning framework for learning Wassertein geodesics, also known as optimal transport geodesics. The training of GeONet requires only pairs of initial and target distributions and not the Wassertein geodesics, which reduces the input data required during the training phase considerably. The main insight here is that the learning problem of the Wassertein geodesic can be made simpler by noting that the optimality conditions in the dynamic formulation of the optimal transport problem is characterized by a coupled PDE system in primal and dual spaces. Once the training phase is over, the inference stage simply requires a forward pass of the network, and therefore is fast enough to be performed in real time. Experiments are performed on synthetic pairs of distributions that are mixture of Gaussians, and on the MNIST dataset.

**Q2-3 Extent To Which Claims Are Supported By Evidence:**

3: Good: the main claims are supported by convincing evidence (in the form of adequate experimental evaluation, proofs, (pseudo-)code, references, assumptions).

**Q2-4 Reproducibility:**

3: Good: key resources (e.g. proofs, code, data) are available and key details (e.g. proofs, experimental setup) are sufficiently well-described for competent researchers to confidently reproduce the main results.

**Q3 Main Strengths:**

Standard optimal transport methods recompute the geodesic for every pair of input distribution pair. The ideas presented in this paper are interesting and potentially of significant impact as the authors use the KKT optimality conditions in the training phase of the neural operator so that the aforementioned recomputation is avoided. The experimental results seem promising.

**Q4 Main Weakness:**

Clearly, the performed experiments are not sufficient to warrant a full advocacy of the mode due to lack of generalization guarantees. The out-of-distribution experiments are very simple with just a pair of Gaussian mixtures.

**Q5 Detailed Comments To The Authors:**

The paper was slightly hard to read and would greatly benefit from a  step-by-step description of the training process. There are several "floating" sections that confused me. For example, the MLP, Fourier feature architecture and entropic regularization paragraphs are not clearly integrated with the rest of Section 3.

Some more experiments to gauge generalization abilities of the proposed method are welcome.

**Q9 Complying With Reviewing Instructions:**

Yes

---

> ### Author Rebuttal · Authors · 2024-04-07
>
> Q4. [More OOD results] To assess the generalization performance, we added a more complicated OOD setting for 1D and 2D Gaussian mixtures. Training data generation remains the same, whose details can be found in Appendix I. Along with the current paper OOD setting, we also highlighted the training setup in reply to *Reviewer yMMr Q4#5 [OOD construction]*.
>
> In the rebuttal, we considered extra OOD testing data distributions (with more perturbation from the training data distribution) as follows.
>
> (i) 1D: 10 Gaussian mixture components (training data have 6 components). We choose means in [1,9], a slightly expanded domain over the training [2,8], and variances in [0.3,0.4] (same as the current OOD setting). A 100-point mesh is used for evaluations and POT solutions with regularization parameter eps=0.0006.
>
> L1 errors (multiplied by 10^{-2} and averaged over 30 testing pairs) are reported as follows.
>
> t=0:         $14.110 \pm 4.336$
>
> t=0.25:    $18.841 \pm 5.963$
>
> t=0.5:      $22.212 \pm 7.317$
>
> t=0.75:    $19.204 \pm 6.142$
>
> t=1:         $13.819 \pm 4.680$
>
> For the definition and interpretation of the L1 error, please refer to our reply to *Reviewer HV3E Q5#3*. We observe that increasing the width of means from [2,8] to [1,9] raises empirical error by about 10%.
>
> (ii) 2D: 16 Gaussian mixture components (training data have 12 components). We choose means in [0.6,4.4]x[0.6,4.4] (training data have means [0.8,4.2]x[0.8,4.2]). 8 variances in each Gaussian are chosen from [0.25,0.3], and 8 variances from [0.65,0.8]. All covariances are chosen from [-0.15,0.15] for each off-diagonal component in each 2x2 covariance matrix. Evaluations were over a 24x24 mesh, the same as used as GeONet branch input.
>
> L1 errors (again multiplied by 10^{-2}) are reported as follows:
>
> t=0:          $10.210 \pm 1.178$
>
> t=0.25:      $9.823 \pm 1.121$
>
> t=0.5:        $9.976 \pm 1.228$
>
> t=0.75:      $9.672 \pm 1.028$
>
> t=1:           $9.923 \pm 0.944$
>
> Currently, we don’t have a rigorous theory on the generalization performance. As we mentioned in “Section 4.6 Limitations”, one future work is to derive a rigorous generalization error bound for accessing the predictive risk of GeONet.
>
>
> Q5. Thanks for encouraging us to improve the presentation. In particular, we apologize for the writing in Sec. 3. We will restructure the presentation and the floating paragraphs in the next version.

---

### Meta-Review · Area_Chair_TPdx · 2024-04-18

This paper proposes a novel method called GeONet, a deep neural operator framework, for learning Wasserstein geodesics. Traditional methods for computing the Wasserstein distance and geodesic between probability measures require mesh-based discretization and suffer from the curse of dimensionality (computational cost exponentially increases with dimensions). GeONet learns the mapping from a pair of initial and terminal probability distributions to the Wasserstein geodesic connecting them. It leverages the KKT optimality conditions from the dynamic formulation of the optimal transport problem. GeONet achieves comparable accuracy to standard solvers on synthetic data (Gaussian mixtures) and the MNIST dataset.

Strengths:

Potentially significant impact due to avoiding geodesic recomputation.
Interesting application of neural operator learning.
Promising experimental results on synthetic data and MNIST.

Weaknesses:

Limited generalization capabilities based on current experiments.
Unclear presentation, especially in Section 3.
Lacks rigorous theoretical foundation for generalization error.
Questionable superiority over existing machine learning methods for geodesics (already fast and accurate).
Needs more justification for why GeONet is superior to POT, especially in lower dimensions (2D, 3D).